# MULTIMODAL GENERATIVE RECOMMENDATION FOR FUSING SEMANTIC AND COLLABORATIVE SIGNALS

## ABSTRACT

Sequential recommender systems rank relevant items by modeling a user's interaction history and computing the inner product between the resulting user representation and stored item embeddings. To avoid the significant memory overhead of storing large item sets, the generative recommendation paradigm instead models each item as a series of discrete semantic codes. Here, the next item is predicted by an autoregressive model that generates the code sequence corresponding to the predicted item. However, despite promising ranking capabilities on small datasets, these methods have yet to surpass traditional sequential recommenders on large item sets, limiting their adoption in the very scenarios they were designed to address. We identify two key limitations underlying the performance deficit of current generative recommendation approaches: 1) Existing methods mostly focus on the text modality for capturing semantics, while real-world data contains richer information spread across multiple modalities, and 2) the fixation on semantic codes neglects the synergy of collaborative and semantic signals. To address these challenges, we propose MSCGRec, a Multimodal Semantic and Collaborative Generative Recommender. MSCGRec incorporates multiple semantic modalities and introduces a novel self-supervised quantization learning approach for images based on the DINO framework. To fuse collaborative and semantic signals, MSCGRec also extracts collaborative features from sequential recommenders and treats them as a separate modality. Finally, we propose constrained sequence learning that restricts the large output space during training to the set of permissible tokens. We empirically demonstrate on three large real-world datasets that MSCGRec outperforms both sequential and generative recommendation baselines, and provide an extensive ablation study to validate the impact of each component.

## 1 INTRODUCTION

Recommender systems play an important role in helping users navigate vast content landscapes by delivering personalized suggestions tailored to their preferences (Aggarwal, 2016). Among these, sequential recommenders have emerged as a powerful approach that explicitly models the temporal order of user-item interactions, capturing evolving user interests over time (Fang et al., 2019). This temporal sensitivity is particularly valuable in domains where the order of interactions carries significant meaning, such as video (Covington et al., 2016; Ni et al., 2023) or e-commerce (Hou et al., 2024). Typically, sequential recommenders learn an embedding for each item based on its co-occurrence with other items, effectively capturing collaborative information. By leveraging sequence modeling techniques such as recurrent neural networks (Hidasi et al., 2016a) or transformers (Kang & McAuley, 2018), these embeddings are processed to predict a new embedding, which is used to infer the next item via approximate nearest neighbor search.

Sequential recommenders face challenges when dealing with large item sets, as the item embeddings can require substantial memory and computational resources, and they often rely solely on collaborative information without incorporating the semantic attributes of items. To alleviate this, TIGER (Rajput et al., 2023) proposes a generative recommendation framework, where each item is encoded as a unique series of semantically meaningful discrete codes. Here, the next item is recommended by generating a code sequence that maps to the predicted item. Semantic codes enable information sharing across similar items, allowing the model to leverage common features. Additionally, by representing items as series of discrete codes, the memory requirements for storing

large collections of items is drastically reduced. Although generative recommenders offer several theoretical advantages, they struggle to outperform traditional sequential models in large item sets (Yang et al., 2025; Lepage et al., 2025), limiting their adoption in the very scenarios they aim to address.

In this work, we propose the Multimodal Semantic and Collaborative Generative Recommender (MSCGRec), a multimodal generative recommendation method capable of leveraging diverse feature modalities to boost its performance on large datasets. MSCGRec seamlessly integrates collaborative features from sequential recommenders into the generative recommendation framework by treating their learned item embedding as a separate modality. As such, MSCGRec retains the beneficial properties of generative recommenders while elevating its performance via the infused sequential recommender knowledge. Furthermore, while previous studies have mainly focused on the text modality (Rajput et al., 2023; Wang et al., 2024a; Zhu et al., 2024), we propose a novel self-supervised quantization learning approach for images that improves the semantic quality of the derived codes. Lastly, to manage the increased complexity of larger item sets and additional modalities, we enhance the training of the next-item predictor by constraining the output space to exclude invalid code sequences, and show that MSCGRec can handle missing modalities on an item level. The main contributions of this work are summarized below:

- We propose a novel multimodal generative recommendation method that seamlessly integrates sequential recommenders.
- We improve the quality of code predictions by *(i)* proposing a self-supervised quantization learning scheme for images that enhances their semantic code quality, and *(ii)* introducing constrained training to incorporate the code structure into training.
- We conduct a thorough empirical evaluation on datasets an order of magnitude larger than previous work, demonstrating MSCGRec's superior performance compared to generative recommendation baselines. Additionally, to the best of our knowledge, we are the first work to showcase a generative recommendation method that beats sequential recommendation baselines at this scale[1].

## 2 RELATED WORK

**Sequential Recommendation**    Sequential recommendation treats the recommendation problem as a sequence of items, where the goal is to find the next item in the series (Wang et al., 2019). Commonly, these methods rank potential next items by the dot product of the predicted item embedding and a learnable lookup table of item embeddings (Pan et al., 2024). Traditionally, such problems have been approached by the Markov Assumption of conditional independence to reduce the complexity (Rendle et al., 2010; Wang et al., 2015). Over time, underlying assumptions have been weakened by modeling the sequence with Convolutional Neural Networks (Tang & Wang, 2018) or Recurrent Neural Networks (Hidasi et al., 2016a;b; Li et al., 2017; Liu et al., 2018). Relatedly, Ma et al. (2019) use a hierarchical gating network to select relevant items and features for predicting the subsequent items.

Since the attention mechanism was introduced in natural language processing (Vaswani et al., 2017), a lot of work in sequential recommendation has started to build on this idea (de Souza Pereira Moreira et al., 2021). Kang & McAuley (2018) pioneered this subfield by using a decoder-only architecture, which was replaced by a bidirectional model in Sun et al. (2019). Zhang et al. (2019) incorporate item attributes alongside item IDs by predicting not only item but also attribute transition patterns. Wang et al. (2023) integrate item attributes in a pre-training stage to enhance the item's representational alignment with these features. Lastly, Zhou et al. (2020) leverage a Self-Supervised Learning framework to capture the intrinsic data similarities.

**Generative Recommendation**    Generative recommendation (Rajput et al., 2023; Sun et al., 2023) is a recent paradigm within sequential recommendation that takes inspiration from the groundbreaking developments in generative language modeling (Brown et al., 2020; OpenAI, 2023; Grattafiori et al., 2024). Instead of representing each item by a unique ID and embedding, items are represented as unique series of discrete codes (Sun et al., 2023). These codes are usually obtained by residual quantization of the item's text (van den Oord et al., 2017; Lee et al., 2022; Huijben et al., 2024), leading to a hierarchical representation (Ward Jr, 1963; Murtagh & Contreras, 2012; Manduchi et al.,

---

[1]We acknowledge concurrent work by Lepage et al. (2025) whose contributions are complementary to ours.

2023). Importantly, instead of predicting the next item's embedding, generative recommendation methods use a sequence-to-sequence model (Raffel et al., 2020) to directly predict the discrete code sequence that corresponds to the next item.

To include information about co-occurring items, Wang et al. (2024a) regularize the codes to be similar to sequential recommendation embeddings. With a similar goal in mind, Zhu et al. (2024) apply a contrastive loss to capture the semantic information of items and their neighborhood relationships. Wang et al. (2024b) model semantics and collaborative information using a two-stream generation architecture with separate decoders. Recent work has also investigated how Large Language Models can be utilized within this framework (Qu et al., 2024; Zheng et al., 2024; Paischer et al., 2025). Instead of trying to improve the code assignment, Yang et al. (2025) integrate ideas from sequential recommendation into the sequence-to-sequence model, while the concurrent work by Lepage et al. (2025) replaces the commonly employed encoder-decoder architecture with separate temporal and depth transformers. Finally Liu et al. (2025) avoid the two-stage approach, by optimizing the item tokenizer during sequence learning.

**Multimodal Generative Recommendation** While many works have concentrated on codifying items' text attributes, more recently, the focus has shifted towards designing methods that can handle the multimodal nature of items (Deldjoo et al., 2024a;b). Taking inspiration from language modeling, one can treat each modality as a separate language and train the sequence model with additional translation-like tasks to encourage a shared vocabulary (Zhai et al., 2025a; Zhu et al., 2025). Alternative approaches use early fusion to encode the multimodal information into a single code sequence with the use of a multimodal foundation model (Zheng et al., 2025) or by a cross-modal contrastive loss (Zhai et al., 2025b). Similarly, Li et al. (2025) use product quantization (Jegou et al., 2010) to merge the codes of multiple modalities into one new code. Lastly, Liu et al. (2024) propose a graph residual quantizer to encode multimodal and collaborative signals into a shared codebook.

## 3 METHOD

With datasets growing in the number of items, modalities, and semantic features, the complexity, but also potential, of the recommendation task increases. Here, we propose MSCGRec, a multimodal generative recommendation method that naturally scales to the growing item space and seamlessly combines semantic with collaborative information. In Figure 1, we provide a schematic overview of the proposed method. In Section 3.1, we introduce the proposed multimodal framework that combines the strengths of multiple semantic modalities and collaborative features. Subsequently, Section 3.2 describes a novel self-supervised quantization method for images that learns semantically meaningful codes. Lastly, Section 3.3 introduces general improvements to the sequence modeling that support the scaling to larger item sets, and are applicable to any method within the generative recommendation field.

### 3.1 MULTIMODAL GENERATIVE RECOMMENDATION

In quantization-based generative recommendation (Rajput et al., 2023), each item is uniquely described by a series of discrete codes $\boldsymbol{c} = [c_1, \ldots, c_L]$. As such, the next item $i$ is obtained by predicting the corresponding sequence of semantic codes $\boldsymbol{c}_i$ based on the item history $\boldsymbol{c}_{<i}$:

$$\mathcal{L}_{rec}^{(i)} = -\log p(\boldsymbol{c}_i|\boldsymbol{c}_1, \ldots \boldsymbol{c}_{i-1}) = -\sum_{l=1}^{L} \log p(c_{i,l}|\boldsymbol{c}_1, \ldots \boldsymbol{c}_{i-1}, \boldsymbol{c}_{i,<l}) \tag{1}$$

These codes are hierarchically ordered such that items that share certain semantics start with the same code sequence. The hierarchical structure is commonly obtained by Residual Quantization (RQ) (Zeghidour et al., 2021). We refer to Appendix B for background on RQ. Traditionally, text has served as the main semantic modality in generative recommendation systems. To boost performance and enable broader generalization across larger, diverse datasets, it is important to integrate capabilities for processing and understanding multiple, heterogeneous modalities. In addition, despite their descriptive capabilities, semantic codes alone can be insufficient for item recommendation, because they do not capture the co-occurrence patterns of items (Yang et al., 2025). As such, various additional losses have been proposed to capture this information (Wang et al., 2024a;b; Yang et al.,

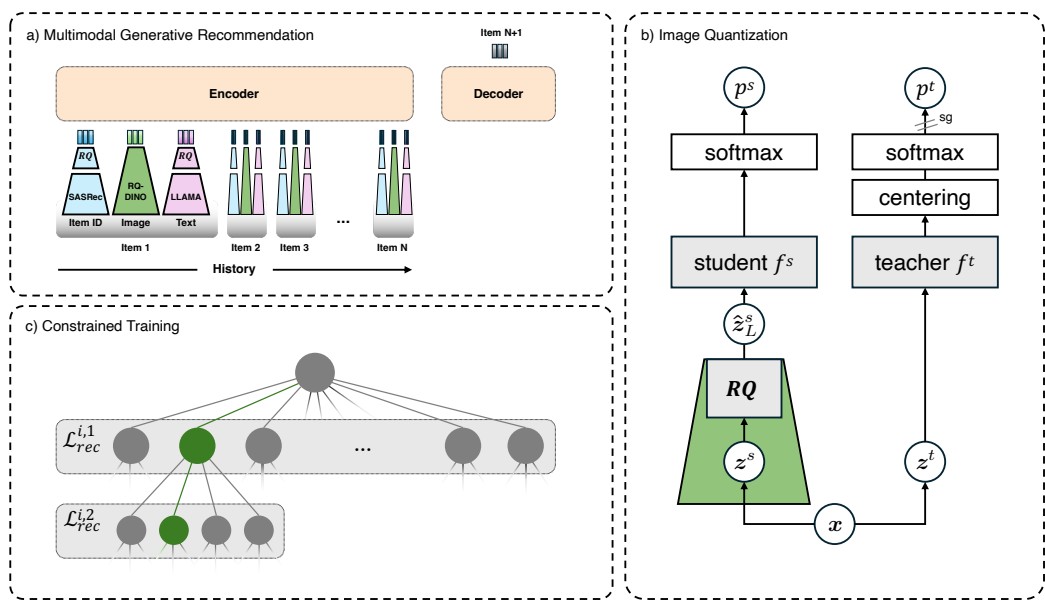

Figure 1: Schematic overview of MSCGRec. (a) Each item in the history is represented by a joint encoding that encompasses all modalities. (b) Images are encoded by self-supervised quantization learning where the student embedding is encoded via residual quantization. (c) Sequence learning is performed by optimizing over permissible codes, where green nodes indicate the codes corresponding to the correct next item.

2025; Liu et al., 2025). In this work, we take a different approach and propose a novel multimodal framework in which collaborative signals can be integrated with semantic features without any additional losses simply by treating the collaborative information as a separate modality. Here, each modality captures different characteristics of the data, which are combined by the sequence learning model. Thus, in contrast to previous work, MSCGRec does not learn a unified encoding but instead leverages the different hierarchical structures of the modalities and empowers the sequence model to extract the structures relevant to the prediction task.

MSCGRec encodes each item as a series of codes of $D$ modalities $\tilde{c}_i = [c_1^{m_1}, \ldots, c_L^{m_1}, \ldots, c_L^{m_D}]_i$. In this work, the utilized semantic modalities are images as described in Section 3.2 and text obtained via standard RQ. Additionally, thanks to MSCGRec's multimodal framework, collaborative features are incorporated by applying RQ on the learned item embedding of a sequential recommender. Notably, we append a separate collision level per modality such that each item's encoding is unique per modality. This allows MSCGRec to leverage the multimodal encoding for enhancing the history's expressiveness, while decoding the next item by a single modality, thereby retaining the standard decoding speed:

$$\mathcal{L}_{rec}^{(i)} = -\log p(c_i^{m_d}|\tilde{c}_1, \ldots \tilde{c}_{i-1}), \tag{2}$$

for a modality $d$. The unimodal decoding arises from opting to stack the codes sequentially instead of concatenating them, as done in Li et al. (2025). At inference, this enables a more effective constrained beam search focused on a single encoding, rather than performing a joint search across multiple hierarchies, which tends to be poorly calibrated.

In real-world multimodal datasets, it is common that not all modalities are available for each item. For example, in PixelRec (Cheng et al., 2023), 30% of items do not have a text description. This demonstrates an additional benefit of retaining modality-specific encodings, as MSCGRec can naturally be extended to handle missing modalities in the user history. The training can be extended by masking a random modality per item with a probability $p$ and replacing the corresponding codes with learnable mask tokens. This optional extension does not have a strong effect on model performance while enabling the model to handle missing modalities.

## 3.2 IMAGE QUANTIZATION

Unimodal generative recommenders have primarily focused on the text modality (Rajput et al., 2023; Wang et al., 2024a; Zhu et al., 2024). As such, it is common to use a pretrained encoder (Ni et al., 2022) to obtain input embeddings, before passing them to the RQ. Since the proposed multimodal framwork allows for the integration of various modalities, we are interested in quantizing images. For the image modality, RQ has been explored in image generation tasks, where raw pixels are used directly as input $\boldsymbol{x}$ (van den Oord et al., 2017; Razavi et al., 2019; Yu et al., 2024; Bachmann et al., 2025). As such, for MSCGRec's multimodal framework, the image quantization's encoder is trained directly on the images. This approach helps mitigate potential domain shift and enables the learning of an embedding space that adheres to the hierarchical structure. In Table 3, we also provide empirical evidence that the usage of a frozen, pre-trained image encoder leads to suboptimal performance. However, while generative image modeling aims to compress the complete image information, the objective of a recommendation system is to extract only the semantically meaningful information. Consequently, we move away from the reconstruction objective, which aims to preserve all information, and instead propose a quantization approach based on Self-Supervised Learning (SSL).

We propose a self-supervised quantization learning approach based on the DINO framework, which is a state-of-the-art image-based SSL method (Caron et al., 2021; Oquab et al., 2024; Siméoni et al., 2025). Unlike CLIP (Radford et al., 2021), DINO does not rely on image-text pairs, making our approach applicable even when a paired text modality is absent or weakly aligned, as is the case in datasets like PixelRec (Cheng et al., 2023). To the best of our knowledge, this is the first work to combine Residual Quantization with the DINO framework. DINO performs self-distillation by training a student model $g^s$ with projection head $f^s$ to match the teacher's output $f^t(g^t(\boldsymbol{x}))$, where the teacher is the exponential moving average of the student's past iterates (He et al., 2020):

$$\mathcal{L}_{DINO} = CE(f^s(\boldsymbol{z}^s), f^t(\boldsymbol{z}^t))); \quad \boldsymbol{z}^s = g^s(\boldsymbol{x}) \ \& \ z^t = g^t(\boldsymbol{x}) \tag{3}$$

We directly incorporate quantization into the DINO framework by applying RQ on the intermediate embedding $\boldsymbol{z}^s$. Importantly, only the student is quantized, thereby nudging the model towards a representation whose quantized approximation retains as much of the teacher's expressiveness as possible. Since the quantization is part of the self-supervised learning framework, the decoder and reconstruction loss that are commonly used for RQ training are no longer required. Instead, the DINO loss provides the learning signal directly.

$$\mathcal{L}_{RQ-DINO} = CE(f^s(\hat{\boldsymbol{z}}_L^s), f^t(\boldsymbol{z}^t))); \quad \hat{\boldsymbol{z}}_L^s = \sum_{l=1}^{L} \boldsymbol{e}_{c_l}^l, \tag{4}$$

where $\boldsymbol{e}_{c_l}^l$ denotes the embedding $\boldsymbol{e}$ at level $l$ that corresponds to the assigned discrete code $c_l$. Putting it all together, we follow the DINOv2 (Oquab et al., 2024) framework with the iBOT (Zhou et al., 2022) and KoLeo loss (Sablayrolles et al., 2019), as well as add a code commitment loss (van den Oord et al., 2017) and update cluster centers via exponential moving average:

$$\mathcal{L}_{RQ-DINO} + \alpha_1 \mathcal{L}_{iBOT} + \alpha_2 \mathcal{L}_{KoLeo} + \alpha_3 \mathcal{L}_{commit} \tag{5}$$

## 3.3 SEQUENCE MODELING

In generative recommendation, each item is assigned a unique series of codes. With bigger datasets, the number of unique code sequences also increases. When analyzing Equation (2) in more detail, it is evident that the loss consists of correctly identifying the next code, but also of identifying the incorrect next codes. To see this, we separate the softmax for item $i$ at code level $l$

$$\mathcal{L}_{rec}^{(i,l)} = -\log \operatorname{softmax}(\boldsymbol{z})_c = -z_c + \log \sum_{c' \in \mathcal{C}} \exp(z_{c'}), \tag{6}$$

where $\boldsymbol{z}$ denotes the predicted logits at position $(i, l)$, and $c$ denotes the correct code within the set of tokens $\mathcal{C}$. As the identification of incorrect codes improves the loss, the model is incentivized to memorize permissible code sequences, meaning which code sequences are assigned to items. However, since the constrained beam search discards impermissible code sequences during inference decoding, memorizing these sequences is unnecessary, and it is only important that the model correctly ranks the permissible codes. This can be considered an instance of shortcut learning (Geirhos et al.,

2020), where the model reduces the loss by learning an unintended behavior. As the number of items increases, the model might allocate a considerable portion of its capacity to memorization, leading to overfitting, especially with a rising number of code collisions. Therefore, restricting MSCGRec's output space to the permissible code sequences aids in focusing on the relevant information.

To enforce that MSCGRec focuses on learning to separate permissible next codes, we adapt the softmax computation such that the normalization factor is only calculated over the set of possible next codes. Formally, we define the tree $\mathcal{T}$, which represents all observed code sequences for items $\mathcal{X}$. Given a sequence of codes $c_{<l}$, the set of permissible next codes is defined as the children of the corresponding node $\mathrm{Ch}(v_{c_{<l}}; \mathcal{T})$. As such, the constrained sequence modeling loss is defined as

$$\mathcal{L}_{rec}^{(i,l)} = -z_c + \log \sum_{c' \in \mathrm{Ch}(v_{c_{<l}}; \mathcal{T})} \exp(z_{c'}), \qquad (7)$$

and this formulation is also used in the constrained beam search to score the beams. The constraint does not add computational overhead to the training, as the prefix tree can be precomputed at the start. By restricting the search space during training to permissible codes, the model learns to focus on differentiating the codes that matter. Beyond the performance improvements shown in Table 3, we also observed that early stopping becomes more useful, because the validation loss now captures predictive performance without being biased by memorization performance. To our knowledge, the proposed constrained sequence modeling is applicable to any generative recommendation method.

Finally, we identified that the relative position embedding in the commonly employed T5 model (Raffel et al., 2020) uses logarithmically spaced bins. This does not adhere to the structure of the codes as their modality and levels can be spaced apart. To address this issue, MSCGRec utilizes two distinct types of relative position embeddings: one that operates across items and another that captures the codes within items. The two position embeddings are summed for the final embedding. We maintain the same number of stored embeddings as before by ensuring that the number of bins across items and the number of within-item bins sum up to the original amount. This novel position embedding enables MSCGRec a more comprehensive understanding of the underlying code structure, as it can explicitly process information of the same modality or hierarchy level across items.

## 4 EXPERIMENTS

In this section, we evaluate MSCGRec by comparing it with sequential and generative recommenders. We introduce the experimental setup in Section 4.1, show the benefits of the proposed method in Section 4.2, and present an ablation study to analyze each component's contribution in Section 4.3.

### 4.1 EXPERIMENTAL SETUP

**Datasets and Metrics**   We conduct our experiments on the Amazon 2023 review dataset (Hou et al., 2024), specifically the subsets "*Beauty and Personal Care*" and "*Sports and Outdoors*". The item sets of these subsets are approximately an order of magnitude larger than the commonly employed subsets of the Amazon 2014 and 2018 editions (McAuley et al., 2015; Ni et al., 2019). Additionally, we study the performance on PixelRec, an image-focused recommendation dataset that provides abstract and semantically rich images (Cheng et al., 2023). Following prior literature (Rendle et al., 2010; Zhang et al., 2019), we preprocess the datasets via 5-core filtering to exclude users and items with fewer than 5 interactions. Additionally, for the Amazon datasets, we remove samples with empty or placeholder images, and deduplicate the items by mapping all items with identical image to a shared id. Train, validation, and test sets are obtained via chronological leave-one-out splitting. For the Amazon datasets, each item per training sequence is used as a separate target, whereas for PixelRec, only the last item is used as target. The maximum item sequence length is set to 20. We provide the dataset statistics after preprocessing in Table 1. To evaluate the recommendation performance, we measure top-K Recall (Recall@K), Normalized Discounted Cumulative Gain (NDCG@K), and Mean Reciprocal Rank (MRR@K) with $K \in \{1, 5, 10\}$.

**Baselines**   We measure the performance of established ID-based sequential recommendation methods, as well as generative recommendation baselines. The sequential recommendation baselines – GRU4Rec (Hidasi et al., 2016a), BERT4Rec (Sun et al., 2019), Caser(Tang & Wang, 2018),

Table 1: Dataset statistics after preprocessing. Subset names are abbreviated.

| Dataset | #Users | #Items | #Interactions | Sparsity |
|---------|--------|--------|---------------|----------|
| Beauty | 724,796 | 203,843 | 6,426,829 | 99.996% |
| Sports | 408,287 | 151,632 | 3,438,255 | 99.994% |
| PixelRec | 8,886,078 | 407,082 | 158,488,653 | 99.996% |

SASRec (Kang & McAuley, 2018), and FDSA (Zhang et al., 2019) – are implemented using the open-source recommendation framework RecBole (Xu et al., 2023), and we refer to Appendix A for more information on the individual methods. Our primary focus is on the comparison with methods that share MSCGRec's generative recommendation framework. TIGER (Rajput et al., 2023) obtains semantic codes by residual quantization of a unimodal embedding. We evaluate TIGER for images and text, denoted by subscript $i$ and $t$, respectively. LETTER (Wang et al., 2024a) incorporates collaborative signals by aligning quantized code embeddings with a sequential recommender's item embedding. We use the LETTER-TIGER variant. CoST (Zhu et al., 2024) proposes a contrastive loss that encourages alignment of semantic embeddings before and after quantization. ETEGRec (Liu et al., 2025) departs from the standard two-step training by cyclically optimizing the sequence encoder and item tokenizer, using alignment losses to ensure that sequence and collaborative item embeddings are aligned. Lastly, MQL4GRec (Zhai et al., 2025a) is a recent multimodal generative recommender that uses modality-alignment losses to translate modalities into a unified language. We implement TIGER and CoST ourselves and use the public codebases for the other methods.

**Implementation Details**   We follow Zhai et al. (2025a) and use LLAMA (Touvron et al., 2023) to extract text embeddings for the Amazon datasets, while using the author-provided text embeddings for PixelRec. We use SASRec's item embedding (Kang & McAuley, 2018) as the collaborative modality. We initialize our image encoder from a DINO-pretrained ViT-S/14 and retain the default hyperparameters for training, apart from reducing the number of small crops to 4 (Oquab et al., 2024), and train for 30 epochs. We retain the loss weights of DINOv2 and set $\alpha_3 = 0.01$ to not interfere too strongly with the representation learning capabilities. To obtain discrete codes for each modality, we individually train a residual quantizer (Zeghidour et al., 2021) with 3 levels, each with 256 entries. For MSCGRec, we directly quantize in the embedding space, without any additional encoder-decoder layers, as we did not observe any performance benefits, which we attribute to the inherent expressiveness of the pretrained models. Following Rajput et al. (2023), we add an additional code level per modality to separate collisions into unique code sequences. We experimented with redistributing collisions into empty leaves, as proposed by Zhai et al. (2025a), but did not observe any performance improvements, which we attribute to our constrained training that restricts the solution space of our additional level. When training for missing modalities, we randomly mask one modality per item in the user history with a probability of 75%. Following prior work (Rajput et al., 2023), we use a T5 (Raffel et al., 2020) encoder-decoder model for sequence modeling and train for 25 epochs with early stopping. We use eight self-attention heads of dimension 64, an MLP size of 2048, a learning rate of 0.002, and train with a batch size of 2048. Based on validation performance, we use the collaborative modality's codes as target codes and unbind the output embedding table to separate it from the unimodal input codes. At inference, we use constrained beam search with 20 beams. Models are trained on four A100 GPUs using PyTorch 2 (Ansel et al., 2024).

## 4.2   RESULTS

In this section, we compare the performance of MSCGRec with sequential recommendation, as well as generative recommendation baselines. Table 2 displays the performance of all methods across a variety of datasets and evaluation metrics. Among the sequential recommendation models, the attention-based SASRec generally achieves the highest performance, whereas the CNN-based Caser model is unable to adapt to the datasets' complexity. Interestingly, SASRec exhibits weaker performance on Recall@1, which may be attributed to calibration issues (Petrov & Macdonald, 2023). For PixelRec, BERT4Rec outperforms SASRec, which could be due to the size of the item set.

Among generative recommendation baselines, MSCGRec consistently achieves superior performance among all datasets and metrics. We observe that no unimodal model based on images (TIGER$_i$), text

Table 2: Performance comparison of sequential and generative recommendation methods. The best-performing method for each row is **bolded** and the runner-up is underlined. $\Delta_{GR}$ indicates MSCGRec's improvement compared to the best generative recommendation and $\Delta_R$ the improvement with respect to all recommendation baselines.

| Dataset | Metrics | Sequential Recommendation | | | | | Generative Recommendation | | | | | | | $\Delta_{GR}$ | $\Delta_R$ |
|---|---|---|---|---|---|---|---|---|---|---|---|---|---|---|---|
| | | GRU4Rec | BERT4Rec | Caser | SASRec | FDSA | TIGER$_i$ | TIGER$_t$ | LETTER | CoST | ETEGRec | MQL4GRec | MSCGRec | | |
| Beauty | Recall@1 | 0.0046 | 0.0042 | 0.0029 | 0.0035 | 0.0050 | 0.0030 | 0.0045 | 0.0053 | 0.0043 | 0.0054 | 0.0048 | **0.0060** | +11.1% | +11.1% |
| | Recall@5 | 0.0155 | 0.0146 | 0.0105 | **0.0204** | 0.0169 | 0.0096 | 0.0148 | 0.0168 | 0.0147 | 0.0182 | 0.0148 | **0.0204** | +12.1% | +0.3% |
| | Recall@10 | 0.0247 | 0.0233 | 0.0174 | **0.0317** | 0.0270 | 0.0147 | 0.0226 | 0.0253 | 0.0231 | 0.0284 | 0.0237 | 0.0315 | +10.9% | - |
| | NDCG@5 | 0.0100 | 0.0094 | 0.0067 | 0.0122 | 0.0110 | 0.0063 | 0.0096 | 0.0111 | 0.0095 | 0.0118 | 0.0098 | **0.0132** | +11.9% | +8.2% |
| | NDCG@10 | 0.0130 | 0.0122 | 0.0089 | 0.0158 | 0.0142 | 0.0079 | 0.0122 | 0.0138 | 0.0122 | 0.0150 | 0.0127 | **0.0168** | +12.0% | +6.3% |
| | MRR@5 | 0.0082 | 0.0077 | 0.0054 | 0.0095 | 0.0090 | 0.0052 | 0.0080 | 0.0091 | 0.0078 | 0.0097 | 0.0082 | **0.0109** | +12.4% | +12.4% |
| | MRR@10 | 0.0095 | 0.0088 | 0.0063 | 0.0110 | 0.0104 | 0.0059 | 0.0090 | 0.0103 | 0.0089 | 0.0111 | 0.0093 | **0.0123** | +10.8% | +10.8% |
| Sports | Recall@1 | 0.0039 | 0.0036 | 0.0026 | 0.0016 | 0.0051 | 0.0025 | 0.0045 | 0.0045 | 0.0044 | 0.0051 | 0.0040 | **0.0053** | +3.9% | +3.9% |
| | Recall@5 | 0.0134 | 0.0124 | 0.0092 | **0.0184** | 0.0170 | 0.0082 | 0.0142 | 0.0141 | 0.0143 | 0.0169 | 0.0123 | 0.0175 | +3.6% | - |
| | Recall@10 | 0.0217 | 0.0197 | 0.0152 | **0.0289** | 0.0269 | 0.0127 | 0.0218 | 0.0212 | 0.0219 | 0.0262 | 0.0206 | 0.0272 | +3.8% | - |
| | NDCG@5 | 0.0087 | 0.0080 | 0.0059 | 0.0103 | 0.0111 | 0.0053 | 0.0094 | 0.0093 | 0.0094 | 0.0110 | 0.0082 | **0.0114** | +3.6% | +2.7% |
| | NDCG@10 | 0.0114 | 0.0103 | 0.0078 | 0.0136 | 0.0143 | 0.0068 | 0.0118 | 0.0116 | 0.0118 | 0.0140 | 0.0108 | **0.0145** | +3.6% | +1.4% |
| | MRR@5 | 0.0072 | 0.0065 | 0.0048 | 0.0076 | 0.0091 | 0.0044 | 0.0078 | 0.0077 | 0.0077 | 0.0091 | 0.0068 | **0.0094** | +3.3% | +3.3% |
| | MRR@10 | 0.0083 | 0.0075 | 0.0056 | 0.0089 | 0.0104 | 0.0050 | 0.0088 | 0.0087 | 0.0087 | 0.0103 | 0.0079 | **0.0106** | +2.9% | +1.9% |
| PixelRec | Recall@1 | 0.0059 | 0.0060 | 0.0034 | 0.0044 | 0.0048 | 0.0006 | 0.0004 | 0.0026 | 0.0002 | 0.0047 | 0.0016 | **0.0066** | +40.4% | +10.0% |
| | Recall@5 | 0.0198 | 0.0212 | 0.0120 | 0.0215 | 0.0172 | 0.0021 | 0.0015 | 0.0091 | 0.0007 | 0.0163 | 0.0063 | **0.0221** | +35.6% | +2.8% |
| | Recall@10 | 0.0303 | 0.0328 | 0.0187 | **0.0335** | 0.0270 | 0.0032 | 0.0026 | 0.0142 | 0.0011 | 0.0250 | 0.0102 | 0.0334 | +33.6% | - |
| | NDCG@5 | 0.0129 | 0.0137 | 0.0077 | 0.0130 | 0.0110 | 0.0014 | 0.0010 | 0.0059 | 0.0004 | 0.0106 | 0.0039 | **0.0144** | +35.8% | +5.1% |
| | NDCG@10 | 0.0163 | 0.0174 | 0.0099 | 0.0169 | 0.0142 | 0.0017 | 0.0013 | 0.0075 | 0.0006 | 0.0133 | 0.0052 | **0.0181** | +36.1% | +4.0% |
| | MRR@5 | 0.0107 | 0.0112 | 0.0063 | 0.0103 | 0.0090 | 0.0010 | 0.0009 | 0.0048 | 0.0003 | 0.0087 | 0.0031 | **0.0119** | +36.8% | +6.3% |
| | MRR@10 | 0.0120 | 0.0127 | 0.0072 | 0.0119 | 0.0103 | 0.0012 | 0.0010 | 0.0055 | 0.0004 | 0.0098 | 0.0037 | **0.0134** | +36.7% | +5.5% |

(TIGER$_t$, CoST), or collaborative signals (ETEGRec) challenges the performance of MSCGRec. This shows the benefit of a multimodal approach and indicates that MSCGRec reliably fuses the information content present in the different modalities. Notably, LETTER and ETEGRec which incorporate collaborative information consistently perform well, with ETEGRec being the runner-up generative recommendation method. However, ETEGRec's current implementation struggles to handle the frequent collisions that occur in large datasets, which were addressed by increasing its capacity with an extra code layer and doubling the number of codes per layer. This highlights the importance of integrating collaborative signals into the generative recommendation framework while avoiding associated drawbacks. Notably, most generative recommendation methods struggle with PixelRec, likely because it contains the biggest item set. In contrast, MSCGRec's effective integration of the image modality and its constrained training over permissible codes results in a significant performance improvement. Lastly, when comparing MSCGRec to MQL4GRec, a state-of-the-art multimodal approach, we note that, aside from not incorporating collaborative information, MQL4GRec processes each modality separately rather than in a unified manner. In contrast, MSCGRec enables the interaction between modalities by jointly passing them as part of the input.

When comparing sequential recommendation methods to generative recommendation approaches, MSCGRec stands out as the only generative method capable of matching or surpassing the performance of the sequential models on these large datasets. Notably, MSCGRec achieves excellent Recall@1, which translates into strong performance on metrics that consider the ranking of items. While sequential models like SASRec excel at modeling user-item interaction sequences, generative recommendation frameworks such as MSCGRec offer unique advantages, most notably the ability to store and operate on discrete codes, rather than high-dimensional embeddings. Although the use of discrete codes can severely restrict the expressiveness of the encoding, our results show that MSCGRec can outperform SASRec, highlighting the value of shared semantic codes. Furthermore, the generated codes correspond directly to the predicted item, which helps avoid the scalability challenges faced by sequential recommenders that rely on traditional atomic IDs and approximate nearest neighbor search. As the first generative recommendation method to surpass sequential recommenders on large datasets, MSCGRec demonstrates that generative recommenders not only offer theoretical advantages over sequential methods, but can also achieve superior performance.

## 4.3 ABLATION STUDY

**Masking & Modality Importance** In Table 3 (a), we show that masked training does not significantly affect model performance, indicating that enhancing MSCGRec with the ability to handle

Table 3: Ablation study of MSCGRec's components and modalities. Component ablations are with respect to MSCGRec, while modality ablations are with respect to MSCGRec with masking. Additionally, we provide an analysis of the performance when only utilizing the image modality.

| Dataset | Metrics | MSCGRec | (a) Component Ablation | | | (b) Modality Ablation | | | (c) Image-Only | |
|---|---|---|---|---|---|---|---|---|---|---|
| | | | w/o Pos. Emb. | w/o Const. Train. | w/ Masking | w/o Img | w/o Text | w/o Coll. | RQ-DINO | DINO |
| Beauty | Recall@10 | 0.0315 | 0.0311 | 0.0291 | 0.0312 | 0.0308 | 0.0299 | 0.0275 | 0.0173 | 0.0158 |
| | NDCG@10 | 0.0168 | 0.0166 | 0.0154 | 0.0166 | 0.0163 | 0.0159 | 0.0146 | 0.0094 | 0.0086 |

missing modalities does not substantially alter its performance. Using this extension, we measure the effect of removing a modality from the input history in Table 3 (b). We observe that MSCGRec's integration of collaborative information is the strongest contributor to performance. Still, even without collaborative features, MSCGRec is better than all other generative recommendation baselines, apart from being slightly beaten by ETEGRec, which bases strongly on the collaborative embeddings. Furthermore, the removal of text or image modalities leads to only a modest reduction in performance. This suggests that MSCGRec learned to leverage the shared information of the semantic modalities, allowing it to maintain robust recommendations even when one is absent. Crucially, these findings underscore the flexibility and resilience of MSCGRec's multimodal framework. The impact of the modality ablation is inherently dataset-dependent, and the observed effects may differ across various datasets and domains, further highlighting the utility and versatility of our approach.

**Image Quantization** MSCGRec's multimodal framework expands semantic modalities beyond text to include images. As such, we propose self-supervised quantization learning, which is also applicable in pure image datasets. To ablate the effect of the proposed image encoding, we provide an image-only analysis of MSCGRec in Table 3 (c). We compare the performance when using the proposed RQ-DINO as encoder with applying RQ post-hoc on a frozen, pretrained DINO model. Evidently, the proposed self-supervised quantization learning provides improvements compared to the common post-hoc approach. This indicates that the self-supervised integration of RQ into the encoder's training aids the quantization learning and extracts the relevant semantics while ignoring the unimportant high frequencies that a reconstruction-based approach would also capture.

**Sequence Learning** The ablation of the constrained training in Table 3 (a) shows the efficacy of the adapted loss function. Restricting the model to only differentiate permissible codes without memorizing unassigned code sequences allows MSCGRec to focus its capacity on modeling the user history. Relatedly, the adapted positional embedding aids the model in understanding the code structure and improves its ability to model relationships between coupled codes of different items. These proposed changes are not specific to MSCGRec and can potentially benefit any model operating within the generative recommendation framework, especially when scaling to larger datasets.

## 5 CONCLUSION

In this work, we proposed MSCGRec, a multimodal generative recommendation method that seamlessly incorporates semantic and collaborative information. MSCGRec encodes images in a novel self-supervised quantization learning framework and jointly processes all modalities to leverage their interactions, thereby merging the benefits of semantic and ID-based approaches. Additionally, we proposed a constrained sequence loss that restricts the search space to the subset of permissible next tokens. Empirically, we showed that MSCGRec achieves superior performance over both generative and sequential recommendation baselines on three large-scale datasets. As such, MSCGRec demonstrates that the generative recommendation paradigm can be used effectively in scenarios involving large item sets, where traditional sequential recommenders can encounter significant storage and computational constraints. Furthermore, we provided an extensive ablation study to highlight the efficacy of each proposed component. Finally, we showed that MSCGRec can deal with missing modalities, which is important for real-world scenarios where the observed semantic modalities can differ per item. Future work could explore the generalization of the proposed self-supervised quantization learning to other modalities, for example using `dino.txt` (Jose et al., 2025). Our findings highlight the versatility of MSCGRec, which we believe to hold significant potential for generative recommendation, paving the way for exciting advancements in recommender systems.

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

# A SEQUENTIAL RECOMMENDATION BASELINES

Here, we provide a short description of each sequential recommendation baseline.

- GRU4Rec (Hidasi et al., 2016a) is an RNN-based sequential recommendation model that uses a customized Gated Recurrent Unit (GRU) to capture user behavior sequences.
- BERT4Rec (Sun et al., 2019) employs bidirectional self-attention with a masked prediction objective to model user preference sequences.
- Caser (Tang & Wang, 2018) utilizes convolutional neural networks with horizontal and vertical filters to capture high-order sequential patterns in user behavior.
- SASRec (Kang & McAuley, 2018) applies a decoder-only self-attention mechanism to model item correlations within user interaction sequences.
- FDSA (Zhang et al., 2019) incorporates feature-level deeper self-attention networks to model both item and feature transition patterns in sequential recommendation.

# B RESIDUAL QUANTIZATION

Residual Quantization (RQ) (Zeghidour et al., 2021) is a technique to compress an embedding into a hierarchical series of discrete codes. For each level of hierarchy, RQ assigns the closest code and subtracts the corresponding code embedding, thereby obtaining a coarse-to-fine sequence of codes. Formally, given an input $\boldsymbol{x}$ and codebooks $\mathcal{C}^l = \{\boldsymbol{e}_k^l\}_{k=1}^{K}$ with $K$ learnable code vectors per level $l \in \{1, \dots, L\}$, RQ computes

$$c_l = \arg\min_{k} \|\boldsymbol{r}_l - \boldsymbol{e}_k^l\|^2 \tag{8}$$

$$\boldsymbol{r}_{l+1} = \boldsymbol{r}_l - \boldsymbol{e}_{c_l}^l, \tag{9}$$

with $\boldsymbol{r}_1 = \text{Encoder}(\boldsymbol{x})$. To capture the semantics of $\boldsymbol{x}$, a reconstruction loss based on $\hat{\boldsymbol{x}} = \text{Decoder}(\sum_{l=1}^{L} \boldsymbol{e}_{c_l}^l)$ is utilized. Additionally, to align the assigned code embeddings $\boldsymbol{e}_{c_l}^l$ with the embedding $\boldsymbol{r}_l$, their $\ell_2$-norm is regularized.

