# OpenReview forum: "Multimodal Generative Recommendation for Fusing Semantic and Collaborative Signals"
_ICLR.cc/2026/Conference — Submitted to ICLR 2026_

### Official Review · Reviewer_tZ7k · 2025-10-29

**Soundness:** 2
**Presentation:** 1
**Contribution:** 2
**Rating:** 2
**Confidence:** 4

**Summary:**

This paper proposes a multimodal generative recommendation framework, MSCGRec, whose core ideas are:

- Discretizing textual, visual, and collaborative signals into modality-specific codes and integrating them in a unified autoregressive framework;
- Introducing RQ-DINO, a self-supervised residual quantization method to learn semantically meaningful visual discrete codes; and
- Adopting constrained vocabularies and constrained beam search during training and inference to avoid wasting computation on impossible code continuations in large candidate spaces.

**Strengths:**

1. RQ-DINO learns visual discrete codes that emphasize product-level semantics (e.g., category, function) rather than low-level visual details, which is more suitable for recommendation tasks.

2. It uses a teacher–student distillation structure with residual quantization, maintaining discriminative capacity under a relatively small codebook size.

3. Ablation results show that RQ-DINO achieves better recommendation performance than the simpler “feature extraction + post-quantization” pipeline.

**Weaknesses:**

1. The motivation stated in the introduction is not well justified. Multimodal representation learning for generative recommendation has already been extensively explored and is widely accepted in GR literature. Likewise, collaborative signals have long been integrated into sequence modeling. Moreover, large-scale industrial systems have already run GR models on datasets with tens of millions of interactions, so this is no longer a novel entry point.

2. Recent and representative GR methods such as [1] and [2] are missing, even though they reflect the latest technical advances in the field. Without them, the empirical comparison lacks persuasiveness.

3. The paper does not compare RQ-DINO against standard residual quantization or VAE/VQ-VAE approaches. Additionally, there is no quantitative report on memory footprint, inference latency, or real-time cost of constrained beam search. These aspects are essential to support claims about scalability.SASRec

4. The SASRec baseline does not appear to incorporate multimodal inputs (text or image features). Hence, part of the observed advantage may stem from richer multimodal information rather than from the generative modeling paradigm itself.

5. The collaborative codes are derived from SASRec’s final item embeddings, but the pipeline description is vague. Details about how these embeddings are trained, whether they are frozen, and whether only training data are used are not provided. This ambiguity raises questions about both computational complexity and the fairness of comparison.

6. The technical presentation is overly abstract and lacks sufficient formalization to ensure reproducibility. For instance, how are the constraints constructed and applied during training and decoding? How is the search space pruned, and how does complexity scale with catalog size? What are the specific hyperparameters (codebook size, number of quantization layers, loss formulation)? Without concrete equations, pseudocode, or implementation details, it is difficult to assess the practical feasibility and replicability of the method.

[1] Actions Speak Louder than Words: Trillion-Parameter Sequential Transducers for Generative Recommendations. ICML 2024.
[2] Contrastive Quantization based Semantic Code for Generative Recommendation. CIKM 2023.

**Questions:**

Please refer to Weaknesses

---

> ### Author Response · Authors · 2025-11-24
>
> We thank the reviewer for their clarifying questions and their interest in our work.
>
> Before going into the specific questions, we want to discuss one important distinction that we believe to be crucial to the assessment of this submission's context and contribution, as well as the raised points.
>
> *There are two fields of research that call themselves Generative Recommendation (GRec)*. One field was introduced by HSTU[1] and tackles the recommendation task using large-scale LLMs. In contrast, the domain we tackle (also called Generative Retrieval), was introduced by TIGER [2] and focuses on quantizing the items into discrete codes with a residual quantizer and subsequently training a generative model (usually a T5) for predicting the codes. *MSCGRec should be allocated within the latter field*, while we interpret the reviewer's comments as inferring the former.
>
> To separate the domains, we mention some key differences: Quantization-based GRec [2] focuses on the discretization of items into semantically meaningful codes via residual quantization and then operates on this discrete input & output domain. This provides a hierarchical, semantic representation of items, strongly reduced storage & computational requirements, cold-start recommendations of new items, and sampling control to generate diverse recommendations using a tunable parameter. On the other hand LLM-based GRec [1] focuses on leveraging huge foundation models and making them computationally feasible for the recommendation task. As such, we deem the two fields not directly comparable. As additional point of reference, we point out that all of our GRec baseline methods [2,3,4,5,6] have not compared to HSTU or related methods as they belong to a different field.
> While we believe that the merging of the two fields would be fruitful, this is outside the scope of our work, and we refer to recent work [7] for an initial attempt at this.
>
> *We apologize for not making the distinction clearer, take full responsibility for it, and will make sure to separate the two lines of research in the manuscript. In light of this distinction, we kindly ask the reviewer to reconsider their sentiment towards our work, as there are fundamental differences between the fields and different points of focus.*
>
> Q1: The motivation stated in the introduction is not well justified.
>
> As we understand it, this comment is based on HSTU-like GRec. Within Quantization-based GRec active research questions are multimodality [5,7,10], integrating collaborative signals[3,6,8], and scaling methods to larger datasets [8,9], all being researched in this year, therefore being highly relevant to this field.
>
> Q2: Missing Related Work [1] & [11]
>
> As discussed above, we do not consider HSTU [1] as a related work. We will make sure to mention this milestone work and clearly mention the differences with MSCGRec to avoid ambiguities. Regarding [11], as we understand it, this work is the initial workshop version of CoST [4], which we compare against. As supporting evidence, please refer to the arxiv versions v1 & v2 [here](https://arxiv.org/abs/2404.14774v1).
>
> Q3: Other quantization approaches and runtime complexity.
>
> The use of RQ-VAE is the standard in the literature [2,3,4,5,6] and has been explored in the initial TIGER paper [2]. As both VAE and VQ-VAE would not pose the memory improvements that RQ-VAE provides -- a key benefit of quantization-based GRec -- we omit this comparison. Regarding runtime complexity and memory footprint, MSCGRec is in line with the other baselines, because thanks to the novel design of MSCGRec, the multimodal input does not affect the length of the output sequence. We refer to Q4 of Reviewer AiRL for more details.

---

> > ### Author Response · Authors · 2025-11-24
> >
> > Q4: The SASRec baseline does not appear to incorporate multimodal inputs (text or image features).
> >
> > We agree with the reviewer that the observed advantages stem partially from the multimodal information. As we consider the multimodal framework of MSCGRec a contribution of ours, we argue that the advantages of leveraging multiple modalities are inherent to our work and therefore provide a valid point of advantage. For example, note that TIGER could also quantize SASRec embeddings (altough we claim the idea for this), however, they are unable to also simultaneously process images and text.
> >
> > Q5: SASRec pipeline
> >
> > To obtain SASRec embeddings, we train a SASRec model using the RecBole library [12] on the identical training data as for MSCGRec's generative model. After that, the embeddings are frozen and treated as fixed input to the quantization module. Utilizing RecBole's default parameters lead to the best validation performance, and we use early stopping on the NDCG@10 validation metric.
> >
> > Q6: Technical presentation
> >
> > We apologize that the Implementation Details paragraph did not contain all relevant parameters and will expand this section in the Appendix of the camera-ready version. Since residual quantization is a core component of most quantization-based GRec methods, we tried to stick to the common choice. As such, to obtain discrete codes for each modality, we individually train a residual quantizer [13, 14] with 3 levels, each with 256 entries. For the loss formulation (apart from RQ-DINO where we refer to Eq. 5), we use the standard RQ losses from [2,14] which consist of a reconstruction loss, commitment loss, and exponential moving average for cluster centers. In line with this, for constrained beam search, we employ the identical techniques as the baseline papers [3,4,5,6] by storing the constraints in a prefix tree that can be computed and recovered efficiently. Then, during training and inference, the set of permissible next codes corresponding to current code sequence is obtained efficiently in O(seq_length) upon which the loss in Eq. 7 is applied. We will make sure to make these connections more explicit.
> >
> > We invite the reviewer to engage in a discussion to make sure we were able to convey the differences between the Generative Recommendation fields and how MSCGRec poses an important contribution to the quantization-based GRec domain. Please let us know if there are any other questions that could further convince the reviewer of our work, and we will be happy to address them.
> >
> >
> > [1] Zhai, Jiaqi, et al. "Actions speak louder than words: trillion-parameter sequential transducers for generative recommendations." _Proceedings of the 41st International Conference on Machine Learning_. 2024.
> >
> > [2] Rajput, Shashank, et al. "Recommender systems with generative retrieval." _Advances in Neural Information Processing Systems_ 36 (2023): 10299-10315.
> >
> > [3] Wang, Wenjie, et al. "Learnable item tokenization for generative recommendation." _Proceedings of the 33rd ACM International Conference on Information and Knowledge Management_. 2024.
> >
> > [4] Zhu, Jieming, et al. "Cost: Contrastive quantization based semantic tokenization for generative recommendation." _Proceedings of the 18th ACM Conference on Recommender Systems_. 2024.
> >
> > [5] Zhai, Jianyang, et al. "Multimodal Quantitative Language for Generative Recommendation." _arXiv preprint arXiv:2504.05314_ (2025).
> >
> > [6] Liu, Enze, et al. "Generative recommender with end-to-end learnable item tokenization." _Proceedings of the 48th International ACM SIGIR Conference on Research and Development in Information Retrieval_. 2025.
> >
> > [7] Zheng, Bowen, et al. "Universal Item Tokenization for Transferable Generative Recommendation." _arXiv preprint arXiv:2504.04405_ (2025).
> >
> > [8] Liu Yang, et al. "Unifying generative and dense retrieval for sequential recommendation." _Trans. Mach. Learn. Res., 2025_
> >
> > [9] Lepage, Simon, Jeremie Mary, and David Picard. "Closing the Performance Gap in Generative Recommenders with Collaborative Tokenization and Efficient Modeling." _arXiv preprint arXiv:2508.14910_ (2025).
> >
> > [10] Zhu, Jing, et al. "Beyond Unimodal Boundaries: Generative Recommendation with Multimodal Semantics." _arXiv preprint arXiv:2503.23333_ (2025).
> >
> > [11] Jin, Mengqun, et al. "Contrastive Quantization based Semantic Code for Generative Recommendation." CIKM 2023.
> >
> > [12] Xu, Lanling, et al. "Towards a more user-friendly and easy-to-use benchmark library for recommender systems." _Proceedings of the 46th International ACM SIGIR Conference on Research and Development in Information Retrieval_. 2023.
> >
> > [13] Zeghidour, Neil, et al. "Soundstream: An end-to-end neural audio codec." _IEEE/ACM Transactions on Audio, Speech, and Language Processing_ 30 (2021): 495-507.
> >
> > [14] Lee, Doyup, et al. "Autoregressive image generation using residual quantization." _Proceedings of the IEEE/CVF conference on computer vision and pattern recognition_. 2022.

---

### Official Review · Reviewer_b46m · 2025-10-31

**Soundness:** 3
**Presentation:** 3
**Contribution:** 3
**Rating:** 6
**Confidence:** 4

**Summary:**

The paper proposes **MSCGRec**, a generative recommender that represents items as discrete code sequences and explicitly **fuses text, image, and collaborative signals**. It is motivated by the memory burden of traditional sequential recommenders on large catalogs and by the observation that prior Gen-Rec models either over-index on text or fail to couple semantics with co-occurrence signals. MSCGRec tackles this by: (1) treating **collaborative features** (from a strong sequential model) as a *separate modality* alongside text and image; (2) introducing **RQ-DINO**, a self-supervised image quantization method aligned to semantics rather than reconstruction; and (3) a **constrained sequence learning** objective that normalizes only over *permissible* code paths, improving scalability. On large, modern datasets (Amazon-2023 Beauty/Sports; PixelRec), the method **matches or surpasses** strong sequential and generative baselines on most metrics, with ablations supporting the role of constrained training, positional encodings, and multimodal quantization. The work is significant as a concrete path toward **scalable generative + multimodal + collaborative** recommendation.

**Strengths:**

1. **High Significance and Impact:** The paper addresses a core limitation of the current generative recommendation (Gen-Rec) paradigm: its failure to outperform strong sequential recommenders on large-scale datasets. By presenting a model that (mostly) matches or exceeds strong baselines like SASRec, this work represents a meaningful step toward the practical adoption of scalable generative models.
2. **Novel and Effective Fusion:** The core idea of treating collaborative features (i.e., item embeddings from a sequential model) as just another "modality" is simple, elegant, and effective. This approach "seamlessly integrates" the power of collaborative filtering, which was a key weakness in previous semantic-only generative models.
3. **Novel Contributions to Scaling:** The paper introduces two valuable techniques for handling large and complex item sets:
   - **Constrained Sequence Learning:** Modifying the softmax loss to normalize *only* over the set of permissible (valid) code sequences is a key contribution. This cleverly prevents the model from wasting capacity on memorizing the vast space of *invalid* code combinations.
   - **RQ-DINO:** The proposed self-supervised quantization method for images, which integrates Residual Quantization directly into the DINO framework, is a novel approach. It avoids a standard reconstruction loss to focus on capturing semantically meaningful information relevant for recommendation.
4. **Strong Empirical Validation:** The authors use three large-scale, modern datasets (Amazon 2023, PixelRec) , which are noted to be "an order of magnitude larger"  than those in prior work. The model is benchmarked against a comprehensive set of strong sequential and generative baselines.

**Weaknesses:**

1. **Lack of Statistical Rigor:** The paper reports no confidence intervals or standard deviations over multiple runs. This is a critical omission, as many of the performance improvements are marginal (e.g., on the Beauty dataset, MSCGRec's R@10 of 0.0315 is *lower* than SASRec's 0.0317 ). Without statistical tests, the claim of "surpassing" sequential models is not adequately supported.
2. **Critical Missing Analysis on Collisions:** The model uses a very small codebook (3 levels, 256 entries each) for very large item sets (e.g., 407k in PixelRec ). This must result in massive code collisions. The paper states it uses an "additional code level... to separate collisions", but provides **no statistics** on this. It is impossible to judge if the model is learning a rich semantic hierarchy or just relying on a "de-facto item ID" at the final level, which would undermine the generative premise.
3. **Missing Key Ablations:**
   - **Decoding Target:** The model's performance is reported when decoding the *collaborative* modality's codes. It is unclear if this performance is due to true multimodal fusion or just "piggybacking" on the powerful pre-trained SASRec embeddings. An ablation showing performance when decoding the text or image codes is essential and missing.
   - **Image SSL Choice:** For the Amazon datasets, which contain paired image-text data, the paper justifies using the image-only DINO  but never compares it against quantizing embeddings from a stronger, multimodal model like CLIP.
4. **Unfair Baseline Comparison (ETEGRec):** The authors state that the ETEGRec baseline's implementation "struggles to handle the frequent collisions" and that they had to modify it by "increasing its capacity". Comparing a tuned, multi-component model against an ad-hoc, modified baseline is unfair and clouds the claim of superiority over this key SOTA generative model.

**Questions:**

1. **Statistical Rigor:** Can you please provide the **mean and standard deviation** over multiple runs for the key results in Table 2? Specifically, are the gains over SASRec statistically significant?
2. **Collision Rate Statistics:** This is critical. Please provide statistics on your codebook. For the 3 semantic/collaborative levels, what percentage of items are unique? What is the average number of items that share the same L1-L3 code prefix *before* the final "collision level"  is added?
3. **Decoding Target Ablation:** What is the model's performance when the decoding target is changed from the collaborative codes  to the **text codes** or the **image codes**? This is essential to prove the model is truly fusing information.
4. **Image Embedding Comparison:** For the Amazon datasets, can you provide a comparison of your RQ-DINO  approach against quantizing embeddings from a pre-trained **multimodal (image-text) model like CLIP**?
5. **Baseline Fairness (ETEGRec):** Can you clarify the exact modifications made to ETEGRec? Can you provide results using a standard, well-tuned implementation to ensure a fair comparison?

---

> ### Author Response · Authors · 2025-11-24
>
> We thank the reviewer for their astute questions and their support of our work. Below is our point-by-point response.
>
> Q1: Standard Deviations
>
> Following the general consensus in the field (e.g. see all of our baselines [1,2,3,4,5,6,7,8,9,10]), we focus on providing point estimates. The variety of metrics, and datasets is intended to serve as an implicit measure of variability. The observed percentual improvements have a similar magnitude than the ones observed in the baseline papers. What we can say is that after having cleaned the code repository, we reran some MSCGRec experiments and only found variability in the fourth decimal point -- with these results we also numerically overtook SASRec Recall@10 in PixelRec and Beauty. Still, we refrained from showing these results in the manuscript to retain scientific integrity and prevent seed-overfitting.
>
> Q2: Collision Rate Statistics
>
> We thank the reviewer for bringing up this point, allowing us to discuss it in more detail. For collaborative embeddings, the number of samples with unique code sequences before the collision level is high, around 95%. For the semantic modalities text and image, this drops to around 70%. We believe this drop can be partially explained by the fact that a substantial amount of Amazon items is near-identical. For example, we observed that oftentimes two separate items differ only by price. To counteract this, we have attempted item deduplication by mapping all items with identical images to a shared id (as mentioned in the paper). Still, we have observed that a large portion of items remains near-identical. Note that this leads to collisions for the text and image modality, but not for the collaborative embeddings, which is exactly what we observe. We leave the investigation of a more thorough deduplication strategy to future work.
> Regarding the doubt of item ID memorization, we refer to our answer to Reviewer VJT9's second question. In short: The TIGER baseline could already perform itemID memorization if it improved performance. Thus, the fact that MSCGRec performs better indicates that the model learns much more than plain memorization.
>
> Q3: Decoding Target Ablation
>
> At this time we are unable to perform this ablation due to computational restraints. To interpret the extent of modality fusion, we refer to Table 3b), where the input modalities are ablated, indicating that no modality is the sole reason for strong performance. In a preliminary experiment during development, we have measured that both image and text targets lead to a decrease of 10% in performance. If the reviewer deems it a necessary additional ablation, we will be more than happy to provide the numbers for the camera-ready version.
> As an alternative approach at measuring modality importance and fusion, we have also computed the Shapley Values [11] (see also Q3 of reviewer AiRL) for each modality. These values are based in game-theory and capture how much each modality (or player) should be attributed to the performance (or payout). It is known in the ML community via SHAP [12] that tries to approximate it for input attributions. As the number of modalities is small, we are able to calculate it exactly by computing MSCGRec's performance for each subset of modalities.
> The resulting Shapley Values for Recall@10 of the Beauty dataset are:
> | Feature (Modality) | Shapley Value ($\phi$) |
> | :--- | :--- |
> | **Image** | 0.0050 |
> | **Text** | 0.0091 |
> | **CF** (Collaborative Filtering) | 0.0123 |
>
> This indicates that while the SASRec embeddings are most important (as also evident by the strong performance of the ETEGRec baseline), the other modalities do play a role in MSCGRec's performance.
>
> Q4: Comparison to CLIP
>
> We believe it would be unfair to compare the image-only RQ-DINO to a multimodal encoder that can leverage both text and images, or displays text-biased image encodings. The intention behind using DINO as backbone is to focus on an image-only model to provide a model that can be applied and finetuned independently of other modalities being present. Additionally, the underlying idea of MSCGRec is that individual modalities should be encoded independently and the generative model should be responsible for fusing their semantics. As such, CLIP would conflict with this view and correspond to an early fusion approach as in [13]. Still, we believe that a RQ-CLIP is an exciting avenue for future work (note that quick preliminary experiments indicated that it is non-trivial to make it work).

---

> > ### Author Response · Authors · 2025-11-24
> >
> > Q5: Baseline comparison to ETEGRec
> >
> > This is a great point that we are happy to elaborate on here, without the manuscript's space constraints. In ETEGRec's code implementation, the collision layer has the same size as the codebooks. As such, if there are more than 256 collision for any leaf, the code will crash. As ETEGRec was not designed to handle datasets of our magnitude, it would exceed this amount every time. As a sidenote in relation to Q2, this also implies a benefit of MSCGRec's encoding of SASRec embeddings, as we did not encounter such stacked collisions. As such, to obtain baseline performance values for ETEGRec, we attempted 3 fixes and presented the best-performing one.
> >
> > (i) We tried increasing the collision layer's size. This lead to strong grouping of items indicative of item memorization, thus, we discarded it.
> >
> > (ii) We left the architecture completely untouched and allowed items to share collision-id's if all 256 collision codes were filled. That is, a leaf's 257th collision was assigned the collision code 1. At inference, we corrected for this duplication by randomly sampling from the items if they shared all codes (note that this adjustment had minor effects on performance).
> >
> > (iii) We increased model capacity s.t. it did not crash. For this, we added an additional level, as well as increased the number of codes per level from 256 to 512. This version performed slightly better than (ii) and hence we display the numbers of this variant in the manuscript.
> >
> > We thank the reviewer for their support in our work and will include the elaborations and experiments in the camera-ready version of the paper. Please let us know if there are any other questions that could further convince the reviewer of our work, and we will be happy to address them.
> >
> >
> > [1] Kang, Wang-Cheng, and Julian McAuley. "Self-attentive sequential recommendation." _2018 IEEE international conference on data mining (ICDM)_. IEEE, 2018.
> >
> > [2] Rajput, Shashank, et al. "Recommender systems with generative retrieval." _Advances in Neural Information Processing Systems_ 36 (2023): 10299-10315.
> >
> > [3] Wang, Wenjie, et al. "Learnable item tokenization for generative recommendation." _Proceedings of the 33rd ACM International Conference on Information and Knowledge Management_. 2024.
> >
> > [4] Zhu, Jieming, et al. "Cost: Contrastive quantization based semantic tokenization for generative recommendation." _Proceedings of the 18th ACM Conference on Recommender Systems_. 2024.
> >
> > [5] Hidasi, Balázs, et al. "Session-based recommendations with recurrent neural networks." _arXiv preprint arXiv:1511.06939_ (2015).
> >
> > [6] Sun, Fei, et al. "BERT4Rec: Sequential recommendation with bidirectional encoder representations from transformer." _Proceedings of the 28th ACM international conference on information and knowledge management_. 2019.
> >
> > [7] Tang, Jiaxi, and Ke Wang. "Personalized top-n sequential recommendation via convolutional sequence embedding." _Proceedings of the eleventh ACM international conference on web search and data mining_. 2018.
> >
> > [8] Zhang, Tingting, et al. "Feature-level deeper self-attention network for sequential recommendation." _IJCAI_. 2019.
> >
> > [9] Zhai, Jianyang, et al. "Multimodal Quantitative Language for Generative Recommendation." _arXiv preprint arXiv:2504.05314_ (2025).
> >
> > [10] Liu, Enze, et al. "Generative recommender with end-to-end learnable item tokenization." _Proceedings of the 48th International ACM SIGIR Conference on Research and Development in Information Retrieval_. 2025.
> >
> > [11] Shapley, Lloyd S. "A value for n-person games." (1953): 307-317.
> >
> > [12] Lundberg, Scott M., and Su-In Lee. "A unified approach to interpreting model predictions." _Advances in neural information processing systems_ 30 (2017).
> >
> > [13] Zhu, Jing, et al. "Beyond Unimodal Boundaries: Generative Recommendation with Multimodal Semantics." _arXiv preprint arXiv:2503.23333_ (2025).

---

### Official Review · Reviewer_AiRL · 2025-10-31

**Soundness:** 2
**Presentation:** 2
**Contribution:** 2
**Rating:** 4
**Confidence:** 4

**Summary:**

MSCGRec is a generative recommender system designed to explain and overcome the performance gap between generative and sequential models on large-scale datasets. Existing generative recommenders often depend too heavily on text semantics and fail to fully exploit collaborative signals. MSCGRec addresses these issues by integrating multiple modalities—text, images, and collaborative features—and treating each as a distinct semantic source. The model introduces several innovations, including self-supervised image quantization with DINO, constrained sequence learning to prevent memorization of invalid codes, and dual position embeddings that capture structural relationships within code sequences.

**Strengths:**

1. This paper advances the emerging generative recommendation paradigm by introducing a principled multimodal framework tha integrates collaborative signals alongside semantic modalities treating them as complementary information sources rather than competing objectives.
2. The paper is well-structured with clear motivation, comprehensive ablation studies, and transparent implementation details. The progression from problem identification to solution design is logical and easy to follow.
3. The experimental evaluation demonstrates that MSCGRec achieves significant performance gains over existing generative methods and, critically, becomes the first generative recommender to surpass traditional sequential baselines on large-scale datasets, validating the paradigm's practical viability.

**Weaknesses:**

1. The paper motivates generative recommendation as a solution to the memory and scalability challenges of traditional sequential models. However, MSCGRec fundamentally depends on SASRec embeddings as the collaborative modality, creating a circular dependency. If the generative approach still requires training a full sequential model as a prerequisite, the claimed advantages (reduced memory footprint, avoiding ANN search) become questionable. This undermines the core value proposition of the generative recommendation paradigm.
2. The methodology lacks principled explanation for why this particular combination of components (DINO, RQ, SASRec embeddings, T5) should work synergistically. The paper reads more as an engineering effort that stacks multiple techniques rather than a theoretically grounded approach. Without formal analysis—such as information-theoretic bounds, convergence guarantees, or at minimum controlled ablations explaining the fusion mechanism—it's unclear whether the gains stem from the multimodal framework itself or simply from leveraging more information sources.
3. Table 3(c) shows image-only performance is poor (Recall@10: 0.0173 vs 0.0315), indicating minimal contribution from the image modality. Why train a custom RQ-DINO encoder from scratch instead of using pre-trained CLIP embeddings? The paper provides no cost-benefit analysis to justify this added complexity given the modest gains.
4. The multi-stage training pipeline and constrained beam search at inference introduce considerable complexity. The paper provides no discussion of total training time, inference latency, or wall-clock comparisons with baselines, making it impossible to assess the practical trade-offs between the modest accuracy gains and substantially increased computational overhead.

**Questions:**

See weaknesses.

---

> ### Author Response · Authors · 2025-11-24
>
> We thank the reviewer for their insightful questions and their interest in our work. Below is our point-by-point response.
>
> Q1: MSCGRec's dependence on SASRec
>
> We thank the reviewer for raising this point, as it allows us to explain in more detail how MSCGRec leverages the strengths of SASRec while avoiding the commonly associated drawbacks. The key insight is that SASRec's embeddings are incorporated into the generative recommendation (GRec) framework. As such, the embeddings are treated like the other modalities by obtaining a quantized encoding in stage 1. After that, the initial embeddings are discarded and only the discrete codes are stored. This means that the memory footprint of a production MSCGRec matches other GRec methods.
> In the generative model of stage two, the items are then decoded via their associated codes. This means also no ANN search is necessary, as the SASRec embeddings are incorporated into the GRec framework. To summarize, the raised uncertainties by the reviewer in fact do not pose a problem, due to the novel inclusion of SASRec embeddings into the GRec framework.
>
> Q2: Attribution of performance gains
>
> Multimodality is a rather new paradigm within Generative Recommendation (GRec) [1,2,3]. To explore the contributions of each component, we have provided Table 3, which among other things quantifies the modality dependence. Additionally, in Table 2 we compare against MQL4GRec, an alternative multimodal framework, to show that MSCGRec beats it consistently. We argue that the inclusion of CF embeddings into the GRec framework also makes a contribution and as such, the leveraging of this additional information source is part of the strengths of the proposed MSCGRec.
>
>
> Q3: On the usage of RQ-DINO
>
> RQ-DINO is compared against its pre-trained counterpart DINO to validate the performance gains, where the R@10 has been 0.0173 compared to 0.0158. While the image modality naturally does not contain the most recommendation-relevant information, we believe that research into better integration of this modality to GRec is warranted. As supporting evidence, we have additionally computed the Shapley Values [4] of each modality. These values are based in game-theory and capture how much each modality (or player) should be attributed to the performance (or payout). It is known in the ML community via SHAP [5] that tries to approximate it for input attributions. As the number of modalities is small, we are able to calculate it exactly by computing MSCGRec's performance for each subset of modalities.
> The resulting Shapley Values for the Recall@10 on the Beauty dataset are:
> | Feature (Modality) | Shapley Value ($\phi$) |
> | :--- | :--- |
> | **Image** | 0.0050 |
> | **Text** | 0.0091 |
> | **CF**  | 0.0123 |
> While the importance ranking of modalities remains unchanged, this analysis shows that images and their analysis still pose value within the GRec field.
>
> Q4: Runtime:
>
> Improving training time has not been an explicit goal of this work, as performance is usually the important factor.
> The RQ-DINO finetuning takes a fraction of the original DINO training, as the pretrained checkpoint provides a good starting point. For the beauty dataset the training took 12 hours, however, we have not investigated early stopping which might shorten the runtime considerably. For the encoder-decoder T5 model, MSCGRec's benefit is that it needs to decode the same number of tokens as previous work. That is, by selecting one decoding modality instead of all M, it avoids a factor M. Still, the encoder's input sequence is scaled by this factor and therefore the computational complexity is increased accordingly.
>
> We thank the reviewer for their comments. Our additional experiments and elaborations will be included in the camera-ready version of the paper. Please let us know if there are any other questions that could further convince the reviewer of our work, and we will be happy to address them.
>
> [1] Zhai, Jianyang, et al. "Multimodal Quantitative Language for Generative Recommendation." _The Thirteenth International Conference on Learning Representations_.
>
> [2] Zhu, Jing, et al. "Beyond Unimodal Boundaries: Generative Recommendation with Multimodal Semantics." _arXiv preprint arXiv:2503.23333_ (2025).
>
> [3] Zheng, Bowen, et al. "Universal Item Tokenization for Transferable Generative Recommendation." _arXiv preprint arXiv:2504.04405_ (2025).
>
> [4] Shapley, Lloyd S. "A value for n-person games." (1953): 307-317.
>
> [5] Lundberg, Scott M., and Su-In Lee. "A unified approach to interpreting model predictions." _Advances in neural information processing systems_ 30 (2017).

---

> ### Comment · Reviewer_AiRL · 2025-11-28
>
> Regarding Weakness 1: The proposed model indeed depends on SASRec during the training stage. This dependency characterizes the approach as a model integration rather than a pure generative recommendation framework. True generative recommendation is designed to replace original ID-based approaches (following the common practice of TIGER [1], which uses textual semantic information to create semantic IDs).
>
> Relying on the proposed framework implies that we must still train a SASRec model, which significantly reduces the necessity and novelty of this paradigm. While integrating ID-based models with generative models will certainly increase performance due to the additional training signals, it eliminates the fundamental advantage of generative recommendation.
>
> Regarding Weakness 2: The authors still have not provided a principled explanation; therefore, my concern remains.
>
> Regarding Weakness 3: The authors did not address my concern regarding why CLIP embeddings were not utilized. Furthermore, the results indicate that the image modality is the least important factor, which exacerbates my concerns regarding the architectural choices.
>
> Regarding Weakness 4: Although efficiency is not the primary focus, the authors should report this information to provide guidance for future research.
>
> Overall, my major concerns remain unresolved. I will maintain my original score.
>
> [1] Rajput, Shashank, et al. "Recommender systems with generative retrieval." Advances in Neural Information Processing Systems 36 (2023): 10299-10315.

---

### Official Review · Reviewer_VJT9 · 2025-11-01

**Soundness:** 3
**Presentation:** 3
**Contribution:** 4
**Rating:** 4
**Confidence:** 5

**Summary:**

This paper proposes MSCGRec, a multimodal generative recommendation framework that unifies text, image, and collaborative filtering (CF) signals into a discrete token space, enabling a generative model to jointly leverage semantic and collaborative knowledge. The method introduces RQ-DINO for semantically stronger visual quantization and a constrained prefix-tree guided decoding strategy to prevent shortcut generation. Experiments on large-scale datasets show that MSCGRec achieves substantial improvements and is the first generative model to surpass strong sequential recommenders such as SASRec, demonstrating both effectiveness and efficiency gains.

**Strengths:**

1. The paper motivates the use of RQ for collaborative embeddings by arguing that CF signals inherently possess multi-level semantics. While the overall performance gains and ablations confirm the usefulness of incorporating CF as a discrete modality, there is no diagnostic analysis demonstrating that different RQ levels actually capture distinct semantic granularity (e.g., global clusters vs. fine-grained item distinctions). A more direct validation—such as layer-wise probing, codebook visualization, or semantic clustering across quantization levels—would substantially strengthen this core claim.

2. The proposed RQ-DINO combines DINO-based self-supervised vision features with residual quantization, yielding more semantically meaningful visual codes and improving generative modeling. The prefix-tree guided generation effectively reduces “memorization of valid token combinations” and forces the model to perform more genuine preference reasoning—an important yet underexplored issue in generative recommenders.

**Weaknesses:**

1. The paper assumes that CF embeddings exhibit hierarchical semantics suitable for residual quantization, yet provides no empirical diagnostics (e.g., layer-wise probing or semantic clustering) validating that RQ indeed captures different levels of collaborative structure. This weakens the core motivation for applying RQ to CF signals.

2. Converting CF embeddings into discrete tokens may unintentionally encode implicit item identity, especially in high-capacity codebooks. This raises the concern that the model’s gains may partially stem from memorizing token-to-item mappings rather than genuinely modeling collaborative structure. Although the multimodal and masked-modality experiments provide some indirect evidence of robustness, a more explicit examination—e.g., shuffled-ID evaluation or unseen-item generalization—would be valuable to confirm that CF tokens do not collapse into disguised item IDs.

3. The prefix-tree guided decoding improves performance and mitigates shortcut generation, as supported by ablation results. However, the paper lacks a deeper discussion of its impact on search space, calibration, and diversity. In particular, it is unclear whether hard constraints reduce the discovery of novel or serendipitous items, or whether soft constraint variants or entropy-aware decoding could achieve a better trade-off between correctness and flexibility.

**Questions:**

Please see the weaknesses above.

1. The motivation for applying RQ to collaborative embeddings relies on the assumption that CF contains multi-level semantics. Could authors provide direct evidence supporting this claim? For example, do different RQ levels correspond to different semantic granularities (e.g., global category vs. fine-grained item distinctions)? Or, Can authors provide codebook visualizations, per-level t-SNE plots, or probing classifiers to show that RQ captures progressively finer collaborative signals rather than redundant noise?

2. Since CF embeddings inherently encode item similarity structure, quantizing them into discrete tokens risks encoding implicit item identity. Could authors provide evidence that the model is not relying on memorized token-to-item mappings? Two possible checks: (1) Train with item IDs randomly permuted but keep CF tokenization identical. If performance remains high, this would indicate identity leakage. (2) Can CF tokens generalize to items not seen in training?

3. Prefix-tree constrained decoding improves correctness, but may reduce diversity or calibration. Could authors provide an analysis comparing on (1) Hard vs. soft constraints. (2) Diversity metrics. (3) Whether constrained decoding biases the model toward popular or well-represented code paths? A deeper study would clarify if this method sacrifices flexibility for accuracy.

---

> ### Author Response · Authors · 2025-11-24
>
> We thank the reviewer for their thorough review and their positive signal in soundness, presentation, and especially contribution. Below is our point-by-point response.
>
> Q1: Could authors provide evidence supporting the claim that CF contains multi-level semantics?
>
> Visualizing the hierarchical structure of CF is difficult because by definition the embeddings capture co-occurrence patterns with other items, which are hard to visualize hierarchically. Instead, we perform a quantitative hierarchical evaluation of the hierarchical clustering that is learned by the RQ-VAE. For this, we follow [1] and compute the dendrogram purity(DP) $\in [0,1]$. In short, this metric measures the purity of the learned tree hierarchy with respect to a chosen ground-truth clustering. For our experiment, we utilize the 117 PixelRec tags as classes of interest. Then, for each pair of datapoints with the same class, DP computes the purity of the least common ancestor, where purity is the percentage of samples with the desired class label within a node. The better a learned tree is able to differentiate the given clustering in the various levels of the hierarchy, the higher the metric will be. Note that DP takes the hierarchical structure of the tree into account, as opposed to many other clustering metrics (Acc, NMI, ARI).
>
> For the learned hierarchy from the CF-based RQ-VAE, we obtain a DP of 0.057, which is similar to the DP for the other modalities. To put this number into context, we perform a pseudo hierarchical clustering where we explicitly control the randomness, with the goal of matching our performance, thereby being able to infer the reduction in randomness that the RQ-VAE provides. As such, the devised pseudoclustering assigns a sample to the correct subtree with probability 1 / k, and otherwise to a different, random subtree. We search for k, whose DP matches ours. Since there are 117 tags, if k=117, it would be random/uninformed clustering, while k=1 would be perfect. We find that for k = 13, the DP obtained is 0.056. Thus, we can conclude that the CF-based hierarchical clustering reduces the assignment variability about *9-fold* compared to random. This indicates in a quantitative manner that the learned hierarchical clustering captures meaningful multi-level semantics.
>
> Q2: Does MSCGRec perform well due to recovering itemIDs?
>
> A cornerstone of generative recommendation (GRec) is the fact that each item has a unique encoding. As such, in each baseline GRec work, and for each modality, the model could theoretically memorize token-to-item mappings. This also means, that the performance of each GRec baseline is lower-bounded (in expectation) by the performance of a memorization-only model that solely relies on token-to-item mappings. Thus, we can for example pick the TIGER$_{i}$ baseline and argue that it's performance must be better or equal than a memorization-only model, as otherwise that's exactly what the model would learn. Since MSCGRec beats TIGER$_i$, we can conclude that it is not due to memorization. Also, as pointed out in W2 by the reviewer, masking of any modality leaves MSCGRec's performance above TIGER$_i$, indicating that it has learned item & collaborative semantics, rather than memorizing the token-to-item mapping.

---

> ### Author Response · Authors · 2025-11-24
>
> Q3: Prefix-tree constrained decoding improves correctness, but does it reduce diversity or calibration?
>
> Instead of an empirical analysis, we propose theoretical arguments in favor of constrained decoding:
>
> (i) Calibration: During training, the change to constrained decoding only affects the set of impermissible children (see Eq. 7). These terms are anyway discarded during the commonly utilized constrained beam search at inference. Importantly, the relative contributions of the permissible children in the training remains unchanged (i.e. the discarded impermissible children only affect the normalizing constant). Therefore, their relative training distribution remains unchanged and as such the model's calibration will also remain unchanged. Thus, in expectation, calibration at inference remains unchanged.
>
> (ii) Diversity: With the same argument as above, we conclude that the diversity within the set of observed items remains unchanged, because the training distribution of the permissible tokens remains unchanged. Furthermore, we argue that with constrained decoding, for new items, the diversity is even increased, rather than reduced.
> Recall that standard decoding approaches lead to permissible token memorization (see e.g. Fig. 6 in TIGER[2]). Thus, when adding a new item that has not been seen during training, its token sequence will be predicted with extremely low probability. In contrast, for MSCGRec, the model learns to capture the codes' underlying semantics without memorization. As such, a new item's codes will be selected with a higher likelihood. To support this claim, we have investigated and observed that at inference, if beam search is not constrained, MSCGRec outputs new token sequences that are not in the training corpus. This indicates that the model generalizes and predicts the desired next-item semantics, rather than creating a mapping to items within the training corpus.
> An intriguing research question that follows from this finding is whether the predicted codes with unconstrained beam search could be used to develop new items to improve the coverage of the item corpus. For example, if one code sequence was predicted for multiple users, the RecSys platform might attempt to create or include an item that corresponds to said sequence.
> We leave this investigation to future work.
>
> Our additional experiments and elaborations will be included in the camera-ready version of the paper. Please let us know if there are any other questions that could further convince the reviewer of our work, and we will be happy to address them.
>
> [1] Kobren, Ari, et al. "A hierarchical algorithm for extreme clustering." _Proceedings of the 23rd ACM SIGKDD international conference on knowledge discovery and data mining_. 2017.
>
> [2] Rajput, Shashank, et al. "Recommender systems with generative retrieval." _Advances in Neural Information Processing Systems_ 36 (2023): 10299-10315.

---

> > ### Comment · Reviewer_VJT9 · 2025-11-28
> >
> > Thanks for the responses from the authors. The authors have solved most my concerns. However, for Q2, I still have some questions.
> >
> > The current response argues that the model is not a pure memorization system. However, our concern is about **partial** identity leakage, not about full memorization. A model can still benefit significantly from disguised item IDs while also learning collaborative structure. Since neither the TIGER comparison nor the modality masking breaks the token–item correspondence, these results do not directly invalidate the identity-leakage hypothesis. We therefore reiterate that **an interventional test** is necessary to convincingly demonstrate that CF tokens do not collapse to surrogate item IDs.

---

### Meta-Review · Area_Chair_ieHL · 2026-01-08

**Summary:**

Reviewers acknowledged the authors' efforts in exploring this topic. However, significant concerns remain unresolved, including methodology details, insights, identity leakage, motivation and novelty, and efficiency. Furthermore, the authors did not update the manuscript to address these issues. In its current state, I believe the paper falls below the expected standard for ICLR and would benefit from another round of revision.

**Reviewer Concerns:**

The author rebuttal addressed only a subset of the concerns raised by the reviewers. The authors did not provide follow-up responses to the additional questions from Reviewers VJT9 and AiRL. Furthermore, the manuscript was not updated to incorporate the points discussed or the clarifications provided during the review process.

Key issues including methodology details, insights, identity leakage, motivation and novelty, and efficiency remain unresolved.

**Reviewer Scores:**

- The authors did not address the follow-up questions raised by Reviewer VJT9, so there is no basis for an increased score from this reviewer.
- Reviewer AiRL explicitly stated that they will maintain their original score of 4.
- Additionally, it appears unlikely that Reviewer tZ7k will raise their score.

Therefore, based on the current reviews, I anticipate that the paper will receive three negative reviews and one positive review, assuming Reviewer b46m maintains their initial positive rating.

---

### Decision · Program_Chairs · 2026-01-26

Reject